# Predicting postoperative visual acuity in epiretinal membrane patients and visualization of the contribution of explanatory variables in a machine learning model

**Akiko Irie-Ota**[1][☯]*, **Yoshitsugu Matsui**[1][☯], **Koki Imai**[2][☯], **Yoko Mase**[1], **Keiichiro Konno**[1], **Taku Sasaki**[1], **Shinichiro Chujo**[1], **Hisashi Matsubara**[1], **Hiroharu Kawanaka**[2], **Mineo Kondo**[1]

**1** Department of Ophthalmology, Mie University Graduate School of Medicine, Tsu, Mie, Japan,
**2** Department of Electrical and Electronic Engineering, Mie University, Tsu, Mie, Japan

☯ These authors contributed equally to this work.
* aota.april@gmail.com

**Data Availability Statement:** To clarify our experimental process, we share detailed information about the source code and execution

## Abstract

### Background

The purpose of this study was to develop a model that can predict the postoperative visual acuity in eyes that had undergone vitrectomy for an epiretinal membrane (ERM). The Light Gradient Boosting Machine (LightGBM) was used to evaluate the accuracy of the prediction and the contribution of the explanatory variables. Two models were designed to predict the postoperative visual acuity in 67 ERM patients. Model 1 used the age, sex, affected eye, axial length, preoperative visual acuity, Govetto's classification stage, and OCT-derived vector information as features to predict the visual acuity at 1, 3, and 6 months postoperatively. Model 2 incorporated the early postoperative visual acuity as an additional variable to predict the visual acuity at 3, and 6 months postoperatively. LightGBM with 100 iterations of 5-fold cross-validation was used to tune the hyperparameters and train the model. This involved addressing multicollinearity and selecting the explanatory variables. The generalized performance of these models was evaluated using the root mean squared error (RMSE) in a 5-fold cross-validation, and the contributions of the explanatory variables were visualized using the average Shapley Additive exPlanations (SHAP) values.

### Results

The RMSEs for the predicted visual acuity of Model 1 were 0.14 ± 0.02 logMAR units at 1 month, 0.12 ± 0.03 logMAR units at 3 months, and 0.13 ± 0.04 logMAR units at 6 months. High SHAP values were observed for the preoperative visual acuity and the ectopic inner foveal layer (EIFL) area with significant and positive correlations across all models. Model 2 that incorporated the postoperative visual acuity was used to predict the visual acuity at 3 and 6 months, and it had superior accuracy with RMSEs of 0.10 ± 0.02 logMAR units at 3

environment via https://doi.org/10.5281/zenodo.11372992.

**Funding:** The author(s) received no specific funding for this work.

**Competing interests:** The authors have declared that no competing interests exist.

months and 0.10 ± 0.04 logMAR units at 6 months. High SHAP values were observed for the postoperative visual acuity in Model 2.

## Conclusion

Predicting the postoperative visual acuity in ERM patients is possible using the preoperative clinical data and OCT images with LightGBM. The contribution of the explanatory variables can be visualized using the SHAP values, and the accuracy of the prediction models improved when the postoperative visual acuity is included as an explanatory variable. Our data-driven machine learning models reveal that preoperative visual acuity and the size of the EIFL significantly influence postoperative visual acuity. Early intervention may be crucial for achieving favorable visual outcomes in eyes with an ERM.

## Introduction

An epiretinal membrane (ERM) is a fibrous proliferative tissue formed on the internal limiting membrane (ILM) of the retina. It commonly occurs in individuals >50-years-of-age and is characterized by vision impairment and metamorphopsia. In severe cases, vitrectomy with ERM peeling is performed. Earlier studies have shown that the factors influencing the postoperative visual acuity include the preoperative visual acuity, age, axial length, retinal thickness, and the morphological status of the inner retinal layers and the photoreceptor outer segments. These findings indicated the importance of the preoperative information in determining the postoperative visual acuity [1, 2].

In recent years, there have been efforts made to predict the postoperative visual acuity using the information of the patients and OCT images by deep learning (DL) methods [3, 4]. However, due to the complex algorithms of the DL systems, it is difficult to create an explainable model that can assess the contribution of each explanatory variable on the postoperative visual acuity. This makes it difficult to use these models as a clinical decision support system (CDSS) for preoperative patients.

To overcome this limitation, we have developed two models to predict the postoperative visual acuity at 1, 3, and 6 months after the vitrectomy in eyes with an ERM using Light Gradient Boosting Machine (LightGBM), a decision tree algorithm within non-linear classifiers. Model 1 makes predictions based solely on the preoperative patient information, while Model 2 incorporates the early postoperative visual acuity as an additional feature.

The purpose of this study was to determine the accuracy of these models in predicting the visual acuity at 1, 3, and 6 months after vitrectomy for an ERM. We also report the findings of the Shapley Additive exPlanations (SHAP) values that assessed the contribution of each explanatory variable. This improved the explanation of the models and visualization of the individual prediction process for each patient.

## Materials and methods

### Study design and participants

We studied 67 eyes of 67 patients who had undergone vitrectomy for an ERM between December 2013 and December 2022 at the Mie University Hospital (Mie Prefecture, Japan). The patients were followed for at least six months postoperatively. The surgeries included stand-alone vitrectomy and surgeries in which cataract surgery was performed with the vitrectomy.

There were also cases in which the ILM was peeled. The exclusion criteria are 1) cases with different degrees of cataract between the two eyes of a single patient, 2) cases with a Snellen visual acuity of less than 20/20 in the fellow eye, 3) cases with ocular disease causing visual impairment other than ERM in the subject eye, 4) poor quality OCT images making it difficult to determine the boundary of the different retinal layers, and 5) OCT images not from the Spectralis® device (Heidelberg Engineering, Heidelberg, Germany). Subjects included in this study were consecutive cases that did not meet any of the exclusion criteria.

## Ethical considerations

The procedures used in this study adhered to the tenets of the Declaration of Helsinki. All protocols were approved by the Ethics Committee of Mie University Hospital (approval number: H2019-219). This study employed an opt-out consent process. The data for this study were analyzed anonymously to ensure participant privacy and confidentiality. The data were accessed between the 20th and 27th of December 2022. The authors had no access to information that could identify individual participants during or after data collection.

## Explanatory variables

For Model 1, the explanatory variables were selected from 22 preoperative features using the variance inflation factor (VIF). These 22 features included the age, sex, affected eye, axial length, preoperative best-corrected visual acuity (BCVA), ERM stage [5], and 16 vector information features identified in the preoperative OCT images that have been reported to be correlated with the postoperative BCVA. For Model 2, the BCVA at 1 month postoperative was added to the features of Model 1 for predicting the 3-month postoperative BCVA. In addition, the BCVA at 1 and 3 months was used to predict the 6-month postoperative BCVA.

## Features extracted from OCT images

Two ophthalmologists agreed on the location of a single pixel of the foveal bulge in the 9 mm vertical and horizontal B-scan OCT images, and annotated this pixel in each image. The annotated images were cropped around the foveal bulge at 3 mm (parafovea region), 1.5 mm (fovea region), and 0.5 mm (foveal avascular zone, FAZ), and necessary annotations were made for each feature (Fig 1). A one-pixel wide line annotation was placed on the inner nuclear layer and inner plexiform layer boundary (INL-IPL), and on the high-intensity line of the retinal pigment epithelium (RPE) in the 3 mm cropped image by the two evaluators. The ratio of the length of the INL-IPL to the RPE was calculated as "the inner retinal irregularity index (IRII) of 3 mm." In the images of the 1.5 mm area, the high-intensity reflection lines of the ILM, outer plexiform layer (OPL), and RPE were identified, and a one-pixel wide line was annotated in each image. The area enclosed by the high-intensity reflection line of the ILM and the high-intensity reflection OPL line in the annotated images was quantified as "the ectopic inner foveal layer (EIFL) area of 1.5 mm". The area between the high-intensity ILM reflection line and the high-intensity RPE reflection line was "the central macular thickness (CMT) of 1.5 mm". Additionally, the area enclosed by the high intensity reflection lines of the ellipsoid zone (EZ) and RPE was identified by the two evaluators and annotated in the region of the 0.5 mm images and quantified as "the photoreceptor outer segment (PROS) area of 0.5 mm". The continuity of the high intensity external limiting membrane (ELM) line and high intensity EZ line was subjectively classified into three categories: defective, discontinuous, or continuous, and quantified as "ELM continuity of 0.5 mm" and "EZ continuity of 0.5 mm". Finally, the images of "EIFL area of 1.5 mm" and "CMT of 1.5 mm" were further cropped to a 0.5 mm area

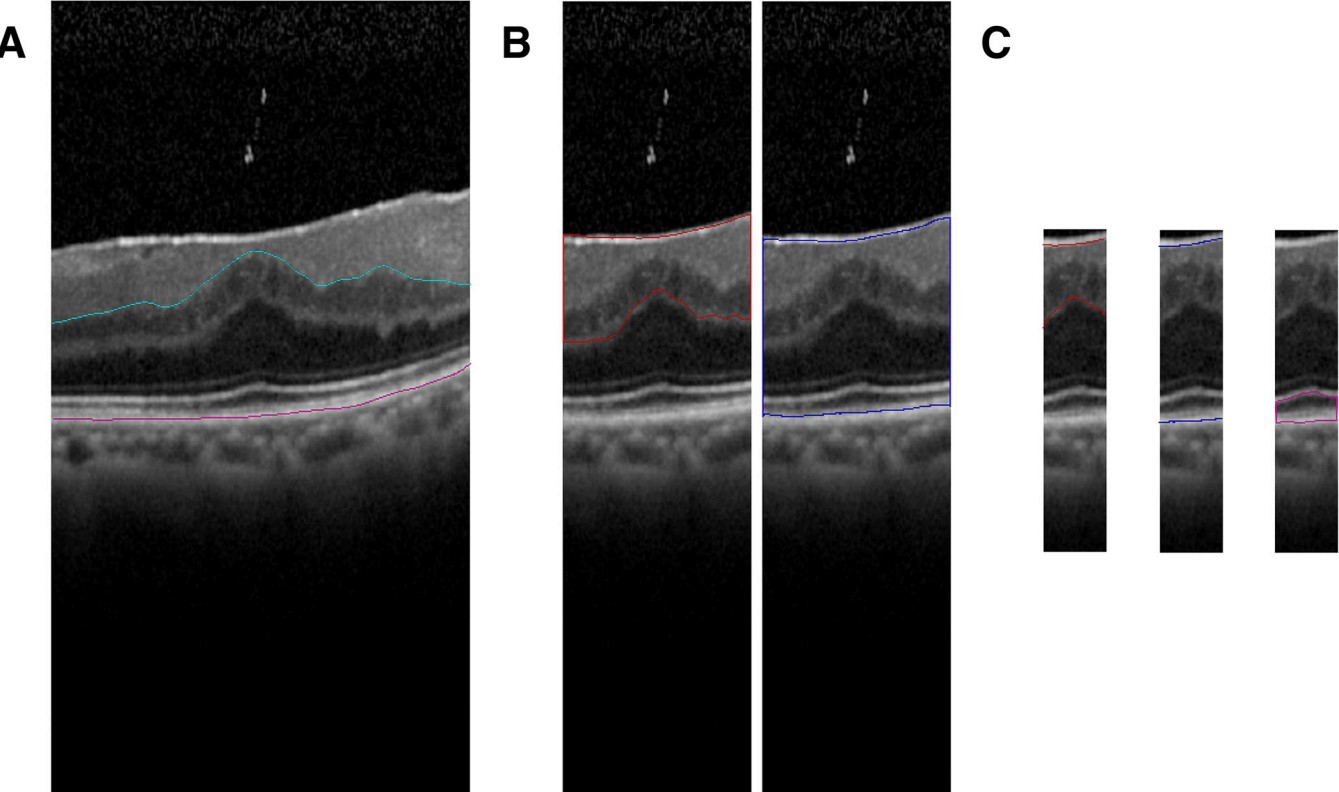

**Fig 1. Annotation of preoperative OCT images.** (A) Optical coherence tomographic (OCT) images that represent the central 3 mm region (parafovea area) in a 9 mm B-scan image at the fixation point. Annotations of the INL-IPL boundary line (cyan line) and retinal pigment epithelium (RPE) high-intensity reflection line (pink line) were made by two ophthalmologists. The ratio of the length of inner nuclear layer and inner plexiform layer boundary (INL-IPL) to the length of RPE was used as the "inner retinal irregularity index (IRII) of 3 mm." (B) OCT image shows a 1.5 mm region (fovea area) centered around the foveal bulge in a B-scan image at the fixation point. The area enclosed by the high-intensity internal limiting membrane (ILM) reflection line and the outer plexiform layer (OPL) high-intensity reflection line (red line) was annotated based on the agreement of two ophthalmologists and used as the "ectopic inner foveal layer (EIFL) area of 1.5 mm." Additionally, the area enclosed by the ILM high-intensity reflection line and the RPE high-intensity reflection line (blue line) was also annotated and used as the "central macular thickness (CMT) of 1.5 mm." (C) The image is a cropped view of a B-scan image centered on the foveal bulge, covering the 0.5 mm region, the foveal avascular zone (FAZ). The area enclosed by the ILM and OPL high-intensity reflection lines was designated as the "EIFL area of 0.5 mm" (red line), and the area enclosed by the ILM and RPE high-intensity reflection lines as the "CMT of 0.5 mm" (blue line). The area enclosed by the EZ high-intensity reflection line and RPE high-intensity reflection line (pink line) was annotated based on the two ophthalmologists and used as the "photoreceptor outer segment (PROS) area of 0.5 mm".

centered on the foveal bulge and were quantified as "EIFL area of 0.5 mm" and "CMT of 0.5 mm" (Fig 1).

## Feature selection

We addressed the multicollinearity by iteratively dropping variables based on their variance inflation factor (VIF). The VIF f for the $j^{th}$ predictor can be calculated using the formula:

$$VIF_j = \frac{1}{1 - R_j^2}$$

where $R^2_j$ is the R-squared value obtained when the $j^{th}$ predictor is regressed against all other predictors. As a general guideline, a VIF higher than 10 indicates significant multicollinearity, suggesting a need for correction. Initially, we ran the model and identified the variable with the highest VIF. This variable was then removed, and the model was rerun. This process of

identifying and dropping the highest VIF variable was repeated until all remaining variables in the model had a VIF of 10 or less.

## Development of machine learning model

An overview of the development of the model is presented in Fig 2. To create the postoperative prediction model for the visual acuity of eyes with an ERM, Light GBM, a decision tree algorithm within non-linear classifiers, was used as the analysis algorithm. Two types of predictive models were developed. Model 1 used the explanatory variables selected from the 22 preoperative features to predict the visual acuity at 1, 3, and 6 months as the target variables. In Model 2, the postoperative visual acuity at 1 month was added as an explanatory variable with the postoperative visual acuity at 3 months as the target variable. Additionally, the postoperative visual acuity at 1 month and 3 months were included as explanatory variables with the postoperative visual acuity at 6 months as the target variable.

For hyperparameter tuning of the predictive models, the entire dataset was divided into 5 parts, and one part was used for validation and the other four for training. A 4-fold cross-validation was performed on the four training folds (Fig 2). After this process, the features with a

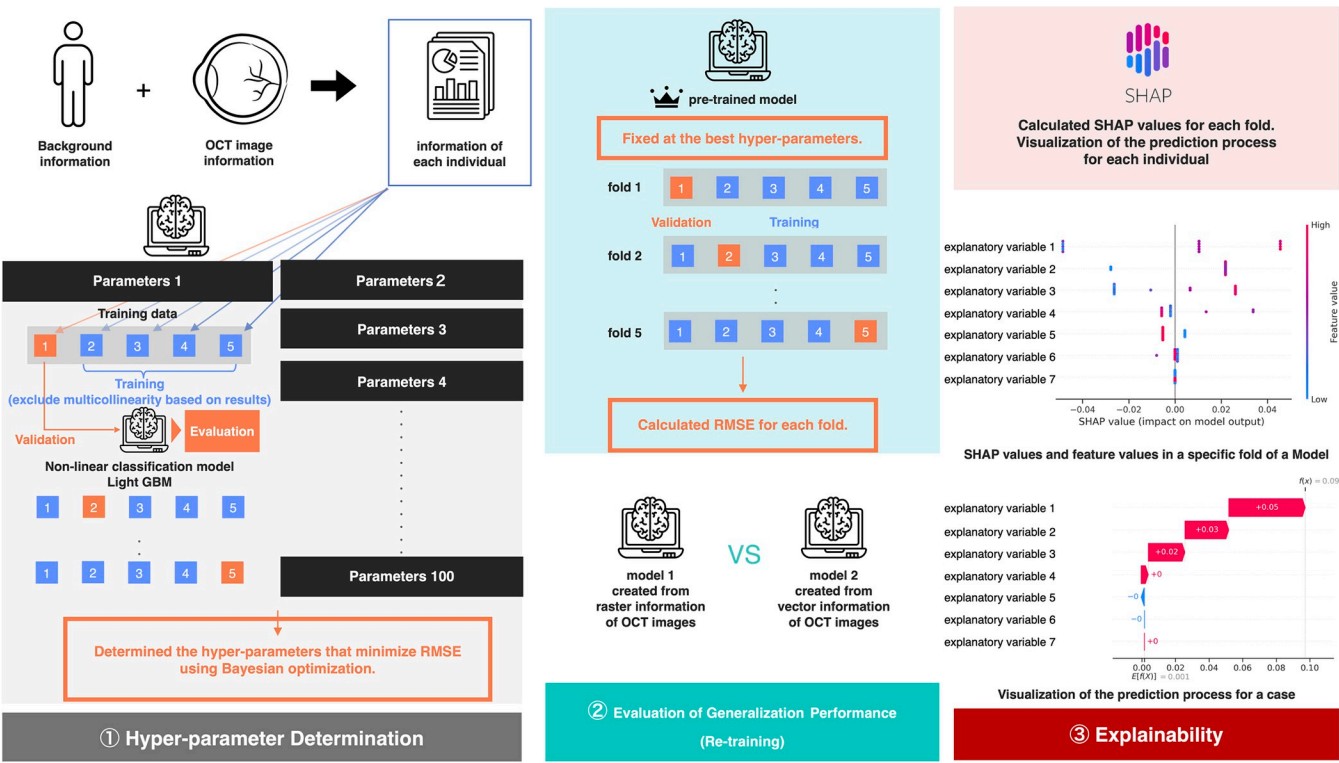

**Fig 2. Overview of the predictive model.** Feature extraction was performed from the preoperative background information and the OCT image data, converting each feature into numerical data. These data were randomly divided into 5 folds with one-fold used for testing and four-folds for training. Then 5-cross validation was conducted, and during this process, multicollinearity of the features was eliminated, and explanatory variables were selected for each model. This random assignment to the 5 folds and 5-cross validation was repeated 100 times. The postoperative visual acuity was predicted using the nonlinear regression algorithm, LightGBM, with prediction accuracy evaluated by the root mean squared error (RMSE). The process involved two stages: the first stage searched for hyperparameters that minimized the RMSE, and the second stage evaluated the generalization performance of both models using the average RMSE with fixed optimal hyperparameters. Finally, the contribution of the explanatory variables in each model's prediction results was visualized using beeswarm plots, and in individual cases using waterfall plots. This figure was created by combining elements from "Human" by Arif Hariyanto, "Eye" by Seaful Muslim, "Data" by AzizGdt, "Brain" by Leo Antho and "Crown" by Kiran Shastry from the Noun Project (https://thenounproject.com). The SHAP logo design is under the MIT License.

VIF greater than 10 ($R^2$ score over 0.9) were excluded to address multicollinearity. Subsequently, the model with the best root mean square error (RMSE) was produced and evaluated using the validation fold. The data allocation to each fold and the 5-cross validations were randomly repeated 100 times to search for hyperparameters that minimized the RMSE in the validation data using Bayesian optimization.

With the optimal hyperparameters fixed, the entire data set was divided into 5 parts again, and 5-cross validations were performed (Fig 2) to evaluate the generalization performance of the model based on the average RMSE of five folds.

### Visualization of contributions of explanatory variables

In evaluating the generalized performance of the models, we used the SHAP values to evaluate the ability of the model to analyze the data which is a feature attribution method based on the Shapley value [6]. The SHAP values of the explanatory variables were calculated for each of the five folds. The Shapley value is represented by the following equation copied from the original SHAP paper:

$$\phi_i = \sum_{S \subseteq F \setminus i} \frac{|S|!(|F| - |S| - 1)!}{|F|!} \left[ f_{S \cup \{i\}}(x_{S \cup \{i\}}) - f_S(x_S) \right]$$

Where $\phi_i$ represents the Shapley value of each feature $i$. $S$ is any subset obtained from the feature set $F$ by removing feature $i$. $f_{S \cup \{i\}}$ and $f_S$ denote the prediction functions of the model with and without including feature $i$.

SHAP is a unified approach that features the importance and offers explanations for both the model structure and the specific features. The SHAP values attributed to each feature change in the expected model prediction when conditioning that feature. They indicate the influence of the prediction value where negative values represent a negative impact and positive values indicate a positive impact on the prediction. Then, their mean absolute values were used to assess the contribution of the explanatory variables in the models.

Additionally, the magnitude and distribution of the contributions of the explanatory variables to the target variable of the model along with the values of each feature, were visualized in beeswarm plots. To interpret the prediction process of each case, the relationship of the SHAP values of each explanatory variable was visualized using waterfall plots.

## Results

### Demographics and clinical characteristics of patients

The background information of the 67 eyes of the patients is summarized in Table 1. The average age of the patients was 69.03 ± 7.39 years with a range of 50 to 81 years. Thirty-six were 36 men (53.7%) and 31 were women (46.3%). On the subject of the stage classification, 7 eyes (10.4%) were at stage 1, 18 eyes (26.9%) at stage 2, 37 eyes (55.2%) at stage 3, and 5 eyes (7.5%) at stage 4. The mean preoperative BCVA was 0.17 ± 0.16 logMAR units, and the postoperative BCVA was 0.08 ± 0.15 logMAR units at 1 month, 0.02 ± 0.12 logMAR units at 3 months, and 0.00 ± 0.13 logMAR units at 6 months. The improvements in the BCVA at each interval were statistically significant ($P < 0.0005$). In the context of surgical procedure, there was one group that underwent vitrectomy alone (one case) and another group that underwent a combination of vitrectomy and cataract surgery (66 cases), and the preoperative BCVA of the vitrectomy alone group was -0.079 logMAR units. Visual acuity at 1, 3, and 6 months postoperatively was 0.222 logMAR units, 0.0 logMAR units, and -0.079 logMAR units, respectively. The mean preoperative BCVA in the group that underwent cataract surgery combined with vitrectomy was

**Table 1. Clinical characteristics and visual outcomes (n = 67).**

| Variables | |
|---|---|
| Mean Age (yrs, range) | 69.03 ± 7.39 (50 to 81) |
| Sex (male, %) | 36 (53.7) |
| Laterality (OD, %) | 28 (41.8) |
| Axial Length (mm, range) | 24.33 ±v1.80 (21.28–30.52) |
| Surgical Procedure(vitrectomy combined with cataract surgery, %) | 66 (98.5) |
| ILM peeling (Yes, %) | 43 (64.2) |
| ERM Stage, 1/2/3/4 | 7 / 18 / 37 / 5 |
| BCVA at baseline (logMAR, range) | 0.17 ± 0.16 (-0.18–0.70) |
| BCVA at 1 month postoperatively (logMAR, range) | 0.084 ± 0.273 (-0.18–0.52) |
| BCVA at 3 months postoperatively (logMAR, range) | 0.019 ± 0.12 (-0.18–0.40) |
| BCVA at 6 months postoperatively (logMAR, range) | 0.0031 ± 0.13 (-0.18–0.52) |

The values are the means ± standard deviations. ILM = inner limiting membrane; BCVA = best-corrected visual acuity; logMAR = logarithm of the minimum angle of resolution.

0.172 ± 0.165 logMAR units. The average postoperative BCVA at 1, 3, and 6 months was 0.082 ± 0.150 logMAR units, 0.019 ± 0.120 logMAR units, and 0.004 ± 0.128 logMAR units, respectively. One patient in the vitrectomy-alone group treated with lens preservation had no worsening of cataracts at 6 months postoperatively. In addition, comparing outcomes of ERM peeling alone (24 patients) versus combined ERM and ILM peeling (43 patients), initial visual acuity improved more in the combined peeling group. Preoperative BCVA was 0.118 logMAR units for ERM alone and 0.196 logMAR units on the average for combined peeling. Visual acuity improvements at 1, 3, and 6 months post-surgery were observed in both groups, with the combined group showing greater improvements. The Kolmogorov-Smirnov (K-S) test indicated significant differences at baseline (p = 0.001) but not at later points (p = 0.299, 0.475, and 0.068).

In addition, the mean and standard deviation of each feature obtained from the preoperative OCT images are shown in Table 2. There were 5 cases of stage 4 of Govetto's classification among the subjects, whereas the INL-IPL boundary was identified in all of them, and the area of IRII (Fig 1A) that we used for our features was not affected.

## Feature selection

In Model 1, six features for the 1-month and five features for the 3 and 6-month prediction models were extracted as explanatory variables through VIF during the hyperparameter adjustments (Table 3). The preoperative visual acuity, affected eye, sex, EIFL area of 0.5 mm, and EZ continuity were consistently selected in all predictions. In Model 2, seven features were selected as explanatory variables for both the 3-month and 6-month prediction of the visual acuity (Table 4). The common features include the preoperative visual acuity, 1-month postoperative visual acuity, sex, affected eye, EIFL area of 0.5 mm and EZ continuity.

## Prediction accuracy

The accuracy of the predictions of each model is presented in Table 5. The RMSE for the 1, 3, and 6 months postoperative visual acuity prediction models were 0.14 ± 0.02 logMAR units, 0.12 ± 0.02 logMAR units, and 0.13 ± 0.03 logMAR units, respectively. For the models that included the postoperative BCVA as an explanatory variable (Model 2), the RMSEs improved

**Table 2. Features of OCT images (n = 67).**

| Variables | |
|---|---|
| IRII of 3mm horizontal image | 1.16 ± 0.13 |
| IRII of 3mm vertical image | 1.28 ± 0.21 |
| EIFL area of 1.5mm horizontal image | 10252.08 ± 3562.04 |
| EIFL area of 1.5mm vertical image | 11048.33 ± 3451.92 |
| CMT of 1.5mm horizontal image | 21550.58 ± 4083.82 |
| CMT of 1.5mm vertical image | 21396.89 ± 4041.07 |
| ELM continuity of 0.5mm horizontal image defective/discontinuous/continuous (%) | 1 (1.5) / 12 (17.9) / 54 (80.6) |
| ELM continuity of 0.5mm vertical image defective/discontinuous/continuous (%) | 1 (1.5) / 13 (19.4) / 53 (79.1) |
| EZ continuity of 0.5mm horizontal image defective/discontinuous/continuous (%) | 1 (1.5) / 17 (25.4) / 49 (73.1) |
| EZ continuity of 0.5mm vertical image defective/discontinuous/continuous (%) | 1 (1.5) / 17 (25.4) / 49 (73.1) |
| EIFL area of 0.5mm horizontal image | 2667.49 ± 1463.12 |
| EIFL area of 0.5mm vertical image | 2705.25 ± 1415.94 |
| CMT of 0.5mm horizontal image | 7188.24 ± 1658.56 |
| CMT of 0.5mm vertical image | 7165.15 ± 1725.79 |
| PROS area of 0.5mm horizontal image | 965.11 ± 216.08 |
| PROS area of 0.5mm vertical image | 965.93 ± 220.20 |

Values are the means ± standard deviations.

IRII = inner retinal irregularity index; EIFL = ectopic inner foveal layer; CMT = central macular thickness;

ELM = external limiting membrane; EZ = ellipsoid zone; PROS = photoreceptor outer segment.

to 0.10 ± 0.02 logMAR units and 0.10 ± 0.03 logMAR units for 3 and 6 months, respectively ($P = 0.03$, $P < 0.01$).

## Visualization of contribution of each explanatory variable

The mean absolute SHAP values of the explanatory variables of each model are presented in Fig 3. In Model 1 for the one-month postoperative period, the mean absolute SHAP values of

**Table 3. Evaluation of multicollinearity in Model 1.**

| Variables | VIF Factor | | |
|---|---|---|---|
| | Post-op 1M | Post-op 3M | Post-op 6M |
| Sex | 2.02 | 1.68 | 1.68 |
| Affected eye | 2.29 | 2.30 | 2.30 |
| logMAR BCVA at baseline | 2.50 | 2.24 | 2.24 |
| EZ continuity of 0.5mm horizontal image | 8.27 | 3.57 | 3.57 |
| EIFL area of 0.5mm horizontal image | | 4.94 | 4.94 |
| EIFL area of 0.5mm vertical image | 5.47 | | |
| PROS area of 0.5mm vertical image | 9.56 | | |

logMAR = logarithm of minimum angle of resolution; BCVA = best-corrected visual acuity; IRII = inner-retinal irregularity index; EIFL = ectopic inner foveal layer; CMT = central macular thickness; ELM = external limiting membrane; EZ = ellipsoid zone; PROS = photoreceptor outer segment; VIF = variance inflation factor.

**Table 4. Evaluation of multicollinearity in Model 2.**

| Variables | VIF Factor | |
|---|---|---|
| | **Post-op 3M** | **Post-op 6M** |
| **Sex** | 1.89 | 1.79 |
| **Affected eye** | 2.78 | 7.47 |
| **logMAR BCVA at baseline** | 2.83 | 2.42 |
| **logMAR BCVA at 1 month postoperatively** | 1.83 | 2.63 |
| **logMAR BCVA at 3 month postoperatively** | | 2.04 |
| **EZ continuity of 0.5mm horizontal image** | 7.77 | |
| **EZ continuity of 0.5mm vertical image** | | 4.38 |
| **EIFL area of 0.5mm horizontal image** | 5.61 | |
| **EIFL area of 0.5mm vertical image** | | 5.66 |
| **PROS area of 0.5mm vertical image** | 9.61 | |

logMAR = logarithm of minimum angle of resolution; BCVA = best-corrected visual acuity; IRII = inner-retinal irregularity index; EIFL = ectopic inner foveal layer; CMT = central macular thickness; ELM = external limiting membrane; EZ = ellipsoid zone; PROS = photoreceptor outer segment; VIF = variance inflation factor.

each explanatory variable (Fig 3A) were in descending order;"EIFL area of 0.5 mm vertical image" was 0.064 logMAR units,"preoperative visual acuity" was 0.027 logMAR units,"PROS area 0.5 mm vertical image" was 0.026 logMAR units,"affected eye" was 0.011 logMAR units,"sex" was 0.0042 logMAR units, and"EZ continuity of 0.5 mm horizontal image" was 0 logMAR units. For the visual acuity at 3 months postoperative time,"preoperative visual acuity" was 0.029,"EIFL area of 0.5 mm horizontal image" was 0.015 logMAR units,"affected eye" was 0.010 logMAR units,"sex" was 0.005 logMAR units, and for the"EZ continuity of 0.5 mm horizontal image" 0 logMAR units. At 6 months postoperative time (Fig 3C),"preoperative visual acuity" was 0.033 logMAR units, "EIFL area of 0.5 mm horizontal image" was 0.014 logMAR units,"sex" was 0.011 logMAR units,"affected eye" was 0.007 logMAR units, and"EZ continuity of 0.5 mm horizontal image" was 0 logMAR units. In Model 2 for the 3 months postoperative time (Fig 3D),"1-month postoperative visual acuity" was 0.055 logMAR units, "preoperative visual acuity" was 0.022 logMAR units,"PROS area of 0.5 mm vertical image" was 0.020 logMAR units,"EIFL area of 0.5 mm horizontal image" was 0.0060 logMAR units,"sex" was 0.0023 logMAR units,"affected eye" was 0.00065 logMAR units, and"EZ continuity of 0.5 mm horizontal image" was 0 logMAR units. At the 6 months postoperative period (Fig 3E),"3-month postoperative visual acuity" was 0.034 logMAR units,"1-month postoperative visual acuity" was 0.028 logMAR units,"preoperative visual acuity" was 0.019 logMAR units,"EIFL area of 0.5 mm horizontal image" was 0.0088 logMAR units,"sex" was 0.0021 logMAR

**Table 5. Prediction results.**

| | Prediction Results (Mean ± SD) | | |
|---|---|---|---|
| **Feature Combinations** | RMSE | | |
| | **1 month** | **3 month** | **6 month** |
| **Model 1** | 0.136 | 0.120 | 0.128 |
| | ±0.015 | ±0.0238 | ±0.0334 |
| **Model 2** | | 0.101 | 0.100 |
| | | ±0.0194 | ±0.0312 |

SD = standard deviations; RMSE = best root mean square error RMSE)

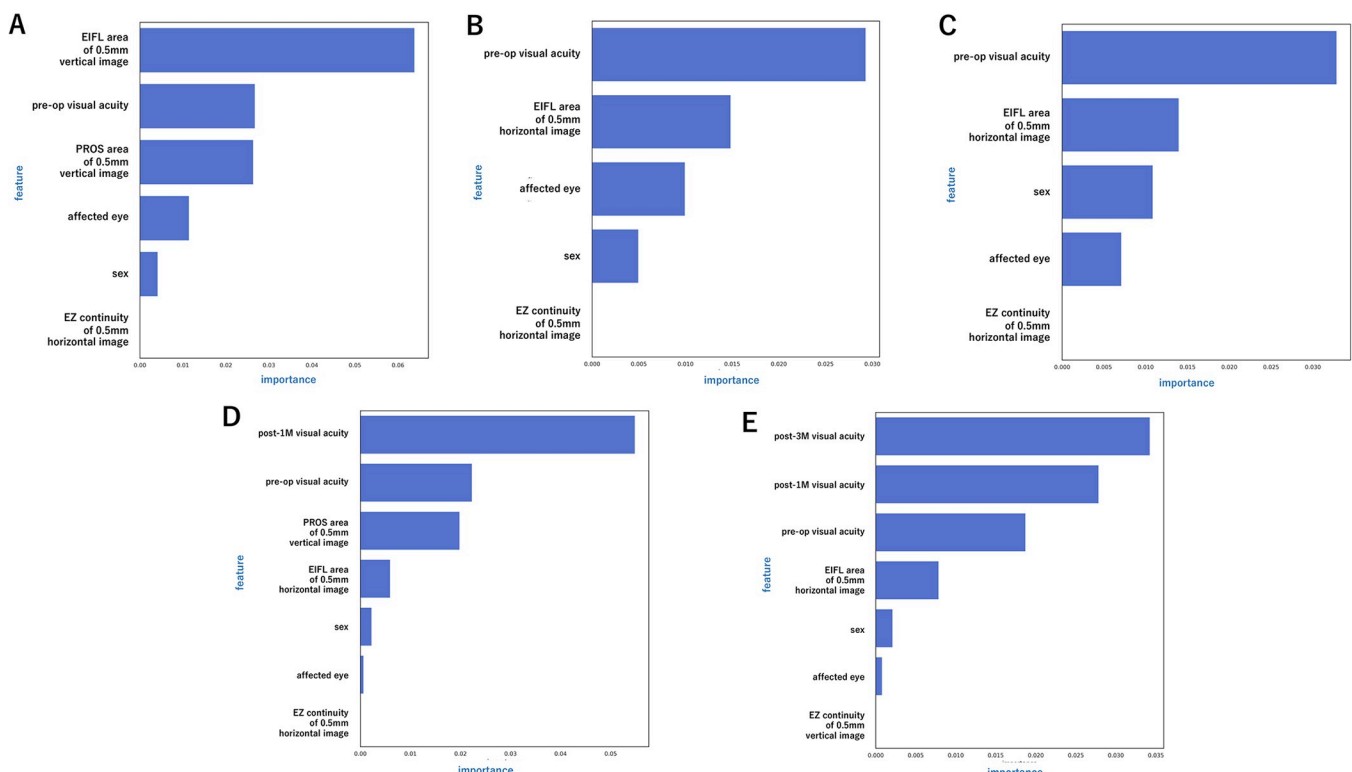

**Fig 3. Mean absolute SHAP values of explanatory variables.** This figure shows the mean absolute Shapley Additive exPlanations (SHAP) values of the explanatory variables in each model. The absolute mean of the SHAP values is represented as a bar graph with only positive values.

units,"affected eye" was 0.00081 logMAR units, and"EZ continuity of 0.5 mm vertical image" was 0 logMAR units.

For the BCVA, beeswarm plots were made for a given fold at each time point of prediction for the two models (Fig 4). In Model 1, the 1-month postoperative beeswarm plot (Fig 4A) showed that the"EIFL area of 0.5 mm vertical image" had the largest horizontal axis distribution of the SHAP values. This indicated that it was a highly influential explanatory variable. For data with larger values, the SHAP values were positive and high, but this trend was not consistent for data with lower values. This indicated a non-linear relationship. Conversely,"-preoperative visual acuity", which had a wide distribution of SHAP values, had a positive correlation in which the lower values led to more negative SHAP values, and larger values led to more positive SHAP values. At 3 months postoperative (Fig 4B),"preoperative visual acuity" had the largest horizontal axis distribution of SHAP values and a significant and positive correlation."EIFL area of 0.5 mm horizontal image" also had a significant effect on the prediction with a positive correlation. At 6 months postoperative time (Fig 4C),"preoperative visual acuity" had the largest distribution, followed by "sex" and"EIFL area of 0.5 mm horizontal image" with both showing a positive correlation with the SHAP values. This was similar to the 3 months postoperative model. For 3 and 6 months postoperative in terms of "sex" and"affected eye", the female patients and left eyes contributed positively towards better outcomes.

Beeswarm plots for Model 2 at 3 and 6 months postoperative are shown in Fig 4D and 4E, respectively. At 3 months,"1-month postoperative visual acuity" and at 6 months,"3-month postoperative visual acuity" followed by"1-month postoperative visual acuity", had the largest SHAP value distributions and were significantly and positively correlated. The"preoperative

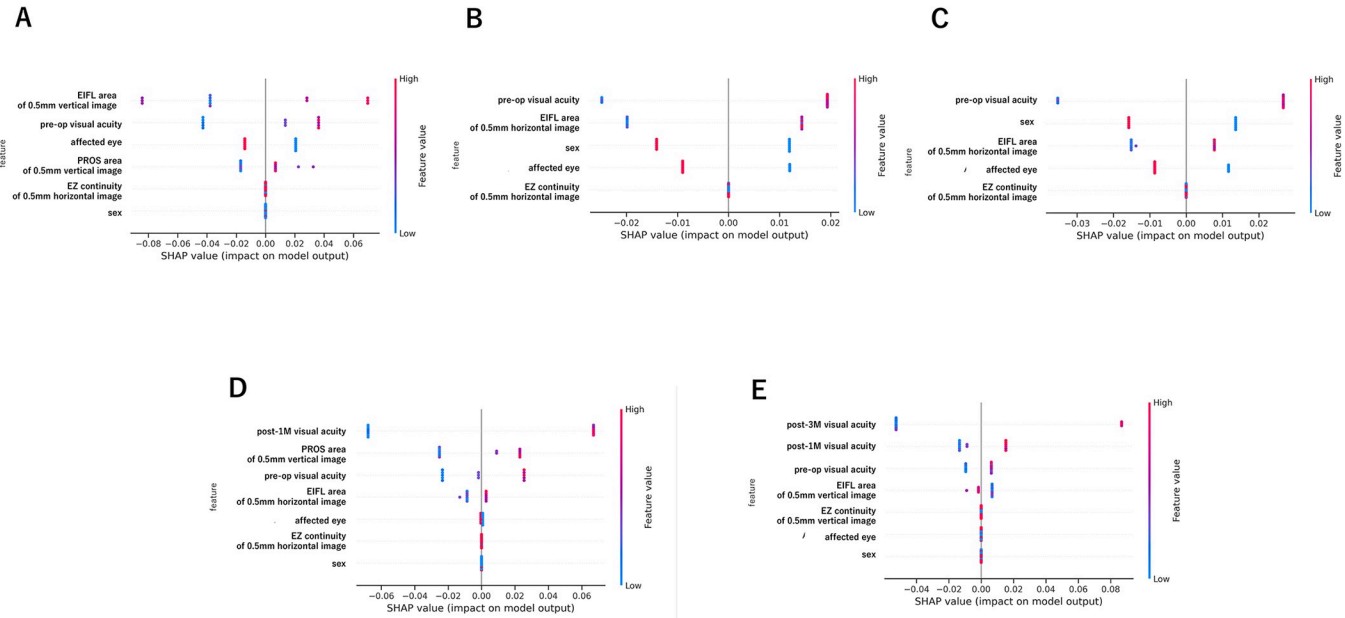

**Fig 4. Visualization of explanatory variable contributions in the models (beeswarm plots).** Each point on the summary plot corresponds to a Shapley Additive exPlanations (SHAP) value for a feature and an instance. Each individual datum of patients with ERM after vitrectomy is represented by a single dot on each feature row. The features on the y-axis are sorted based on their importance; color shows the feature value from low (blue) to high (red). The SHAP value is displayed on the x-axis, with the minus values on the left side show a negative impact and the plus values on the right side show a positive impact. (A) The beeswarm plot for a specific fold of the 1-month postoperative visual acuity prediction in Model 1. (B) Beeswarm plot for a specific fold of the 3-month postoperative visual acuity prediction in Model 1. (C) Beeswarm plot for a specific fold of the 6-month postoperative visual acuity prediction in Model 1. (D) Beeswarm plot for a specific fold of the 3-month postoperative visual acuity prediction in Model 2. (E) Beeswarm plot for a specific fold of the 6-month postoperative visual acuity prediction in Model 2.

visual acuity" also had a positive correlation. In the OCT image features at 3 months postoperative time,"PROS area of 0.5 mm vertical image" and"EIFL area of 0.5 mm horizontal image"," and at 6 months postoperative,"EIFL area of 0.5 mm vertical image" were significantly and positively correlated.

## Visualization of individual prediction processes

The individual postoperative visual acuity prediction processes for each case are visualized in Fig 5. E[f(x)] represents the average visual acuity in logMAR units of the model's predictions for the training data, which served as the baseline. Against this baseline, the contribution of each explanatory variable to the individual output, f(x), of the cases, the postoperative visual acuity in logMAR units was added in the order of increasing SHAP values and direction. Fig 5 presents cases of both Model 1 and Model 2 where the predicted visual acuity is worse and better than the baseline values, respectively.

## Discussion

OCT has simplified the diagnosis of ERM but there are no standardized criteria for the optimal time for vitrectomy to treat an ERM. One reason for this is the difficulty in assessing the relationship between preoperative information and postoperative visual acuity quantitatively. One possible solution might be the use of an explainable machine learning-based CDSS which would enable a more objective and convincing determination of the optimal time. Using the Explainable Artificial Intelligence (XAI) in CDSS can enhance the reliability of decision-making, generate hypotheses about causal relationships, and increase both the system's

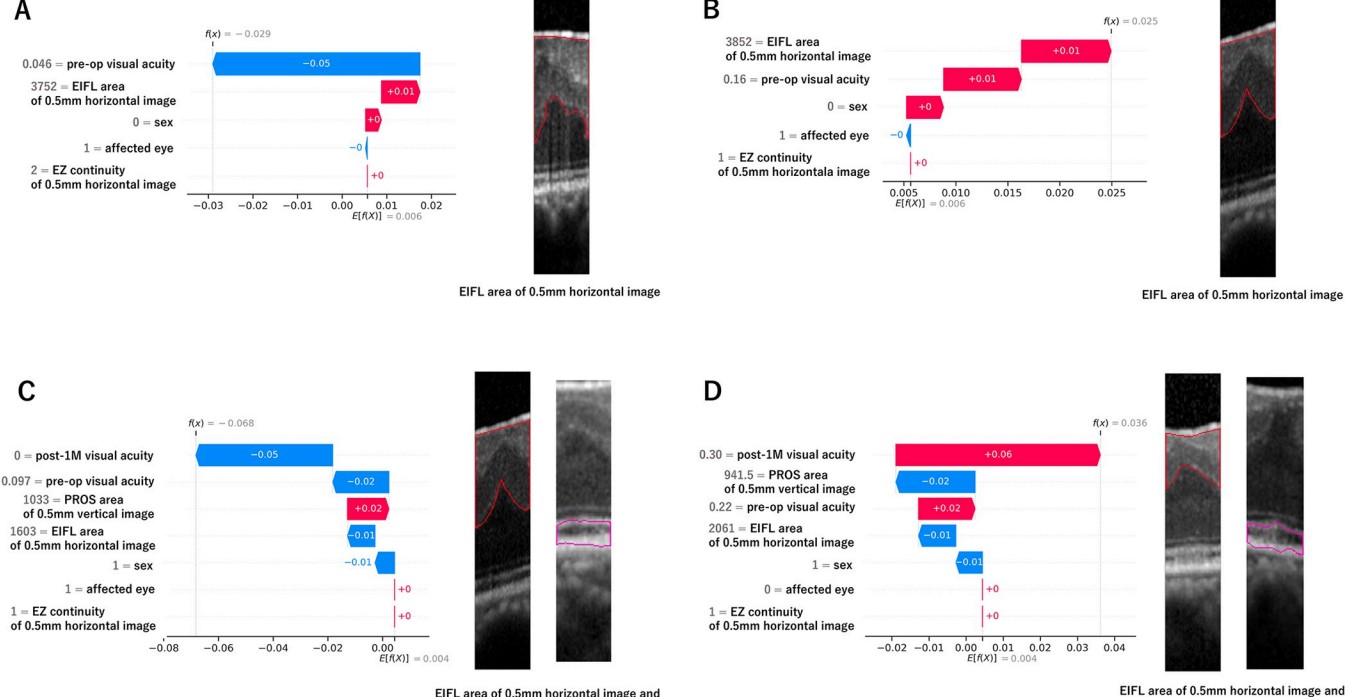

**Fig 5. Visualization of contributions of explanatory variables in individual cases (waterfall plot).** (A) The predictive process for a specific case for the 3-month postoperative visual acuity prediction using Model 1. The process begins with the baseline average visual acuity in logMAR units E[f(X)] from the model's predictions on the training data, to which Shapley Additive exPlanations (SHAP) values of smaller magnitude (indicating lesser influence) are sequentially added. The individual output, f(x), for this case represents the postoperative visual acuity in logMAR units. This case exemplifies an instance where the postoperative visual acuity is better than the baseline due to the contribution of the explanatory variables. In this case, starting from the baseline value E[f(X)] = 0.01, "EZ continuity of 0.5 mm horizontal image" (feature value 2, continuous) and "affected eye" (feature value 0, right) make no contribution. This is followed by a slight positive contribution by "sex" (feature value 0, male), "EIFL area of 0.5 mm horizontal image" (feature value 3752) contributes a negative value of 0.01, and "preoperative visual acuity" (feature value 0.46) contributes a negative value of 0.05, resulting in a predicted outcome f(x) = -0.03. (B) Prediction at 3 months by Model 1 where the postoperative visual acuity is worse than the baseline because of the explanatory variables. (C) and (D): Model 2 show the prediction processes for different cases for the 3-month postoperative visual acuity. C is a case with better-than-baseline visual acuity, while D is a case with worse-than-baseline acuity, both due to the influence of the explanatory variables.

acceptability and trustworthiness. In fact, recent efforts have been made to create models more acceptable and reliable for clinicians by explaining them with game-theoretical XAI methods such as SHAP [7]. Therefore, we attempted to predict the postoperative visual acuity after removal of an ERM by vitrectomy using a nonlinear regression machine learning model. We also simultaneously visualized the contributing factors and prediction process in the models created.

Our machine learning models were created using LightGBM [8]. During the development of the models, explanatory variables were selected from the features obtained during hyper-parameter tuning using cross-validations. Multicollinearity was addressed using the VIF which excluded variables with strong correlations.

General multivariate analysis methods are capable of evaluating the relationships between numerous explanatory variables and a target variable. However, they assume that each variable is independent, and when dealing with variables that have a nonlinear higher-order variable interactions, it is necessary to add interaction terms. In the case of extracting features from OCT images, even explanatory variables representing distinct retinal layers cannot be assumed to be independent of interactions with each other. This then makes these features not suitable for analysis with traditional multivariate methods that do not account for such interactions.

However, nonlinear classifiers such as LightGBM can learn nonlinear relationships and higher-order interactions from data which reduce the need to explicitly add interaction terms to the model. Moreover, LightGBM incorporates regularization methods to prevent overfitting to control the complexity of the interactions. Thus, it can analyze a large number of variables and determine their interactions. Our machine learning model is noteworthy not only for the prognosis but also as a method for statistical analyses.

The importance of each feature was calculated by the averaged absolute SHAP values of the entire independent test set for each feature to help users gain an insight of the overall behavior of the model in our models. We also used the SHAP values of a particular instance to explain how each feature and its value contributed to the predicted postoperative visual acuity.

General explanations for each model structure were represented in the beeswarm plots, and the specific feature explanations in the waterfall plots. These plots helped interpret the results of each prediction model and the predictions for individual patients.

The accuracy of the predictions of our postoperative visual acuity model was comparable to the RMSE of the reported DL models for post-ERM surgery [3, 4]. Although there is some complexity involved in creating features from OCT images, as opposed to using images directly in the DL, the ability to achieve predictions with a comparable RMSE is noteworthy. Furthermore, a major advantage of our model is its ability to visualize the extent of the impact of each explanatory variable on the predictions. This visualization may make our model more useful by offering an enhanced understanding of the predictive factors on the postoperative visual acuity following ERM surgery.

In Model 1, the preoperative visual acuity had the highest SHAP value among the different information about the patient. The beeswarm plot also showed a positive correlation between the preoperative visual acuity and the postoperative predicted visual acuity. This reaffirms that surgery performed at a time when the preoperative visual acuity is good can lead to a better postoperative visual acuity.

In Model 2, which added the 1 and 3 months postoperative visual acuity information ranked high in the SHAP values. This indicated that the postoperative visual acuities had a significant impact on the predictions. The beeswarm plot showed a positive correlation between the postoperative visual acuity and the predicted visual acuity. This is consistent with the tendency that the long-term postoperative visual acuity in patients following ERM removal by vitrectomy continues to improve [9]. Additionally, the prediction accuracy was better in Model 2 than in Model 1 with a significant reduction in the RMSE. This indicated that the postoperative visual acuity is important for more accurate predictions.

Among the OCT features, the area of the EIFL within the 0.5 mm area of the foveal bulge had higher SHAP values. The beeswarm plot showed a significant and positive correlation of EIFL with the postoperative predicted visual acuity. We adopted the OCT features such as the distance and area as vector data. The presence of EIFL as part of Govetto's classification is already known to be an important factor in the prognosis of the postoperative visual acuity [5]. Our model showed that a larger EIFL area within the 0.5 mm range had a worse predicted visual acuity which suggests that early intervention in the stages of Govetto's classification positively impacts the visual prognosis [10].

On the other hand, the region we defined as the EIFL is not exactly the same as the EIFL defined by Govetto. We evaluated the inner retinal layers present in the FAZ as a continuously varying quantity. This included both the normal inner layer structure and EIFL, with larger values corresponding to an increase in the EIFL. This approach could potentially offer a more accurate assessment of the impact of EIFL on the prognosis rather than an evaluation based solely on the stage of the disease.

In the patients' background information, sex emerged as a significant explanatory variable with females positively influencing a better postoperative visual acuity. This agreed with other reports indicating that males have an adverse prognostic factor for post-ERM surgical outcomes [11]. A possible explanation for this could be the differential roles of the sex hormones. The androgens are known to promote inflammation whereas estrogens have been associated with the resolution of inflammation and facilitation of tissue repair [12, 13]. However, an earlier study reported no statistically significant impact of sex on the postoperative visual acuity after ERM surgery [1]. These contradictory findings indicate a need for further research to determine the influence of sex on the visual acuities in postsurgical ERM patients.

There are limitations in this study. An important limitation was its retrospective nature and the small sample size. Because our experiment used a data-driven model, increasing the sample size is crucial. As the research is based on a single institution, performing a multi-center prospective studies in the future will be essential to verify the reproducibility of our findings.

In conclusion, we have developed a model for predicting the postoperative visual acuity after vitrectomy for an ERM. We use a nonlinear algorithm-based machine learning model and achieved moderate success. By calculating the SHAP values, we were able to examine the contribution of each explanatory variable making the model more interpretable and demonstrated the potential of XAI in CDSS. Despite being data-driven, our machine learning models indicated that the preoperative visual acuity and the area of EIFL significantly influenced the postoperative visual acuity. This indicates that early intervention is critical in achieving favorable visual outcomes in eyes with an ERM.

## Acknowledgments

We thank Professor Emeritus Duco I. Hamasaki of the Bascom Palmer Eye Institute of the University of Miami (Miami, FL, USA) for critical discussion and final manuscript editing.

## Author Contributions

**Conceptualization:** Akiko Irie-Ota, Yoshitsugu Matsui, Yoko Mase, Shinichiro Chujo.

**Data curation:** Yoshitsugu Matsui, Keiichiro Konno, Taku Sasaki, Shinichiro Chujo, Hisashi Matsubara.

**Formal analysis:** Koki Imai.

**Software:** Koki Imai.

**Supervision:** Yoshitsugu Matsui, Hiroharu Kawanaka, Mineo Kondo.

**Writing – original draft:** Akiko Irie-Ota.

**Writing – review & editing:** Yoshitsugu Matsui, Yoko Mase, Mineo Kondo.

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
