## [Decision Letter · Decision Letter 0]

13 Feb 2024

PONE-D-23-41738Predicting Postoperative Visual Acuity in Epiretinal Membrane Patients and Visualization of the Contribution of Explanatory Variables in a Machine Learning ModelPLOS ONE

Dear Dr. Irie-Ota,

Thank you for submitting your manuscript to PLOS ONE. After careful consideration, we feel that it has merit but does not fully meet PLOS ONE’s publication criteria as it currently stands. Therefore, we invite you to submit a revised version of the manuscript that addresses the points raised during the review process.

We look forward to receiving your revised manuscript.

Kind regards,

Tatsuya Inoue

Academic Editor

PLOS ONE

4. We note that Figure 2 in your submission contain copyrighted images. All PLOS content is published under the Creative Commons Attribution License (CC BY 4.0), which means that the manuscript, images, and Supporting Information files will be freely available online, and any third party is permitted to access, download, copy, distribute, and use these materials in any way, even commercially, with proper attribution. For more information, see our copyright guidelines: http://journals.plos.org/plosone/s/licenses-and-copyright.

1. You may seek permission from the original copyright holder of Figure 2 to publish the content specifically under the CC BY 4.0 license.

2. If you are unable to obtain permission from the original copyright holder to publish these figures under the CC BY 4.0 license or if the copyright holder’s requirements are incompatible with the CC BY 4.0 license, please either i) remove the figure or ii) supply a replacement figure that complies with the CC BY 4.0 license. Please check copyright information on all replacement figures and update the figure caption with source information. If applicable, please specify in the figure caption text when a figure is similar but not identical to the original image and is therefore for illustrative purposes.

Reviewers' comments:

Reviewer's Responses to Questions

**Comments to the Author**

1. Is the manuscript technically sound, and do the data support the conclusions?

Reviewer #1: Yes

Reviewer #2: Yes

2. Has the statistical analysis been performed appropriately and rigorously? 

Reviewer #1: Yes

Reviewer #2: Yes

3. Have the authors made all data underlying the findings in their manuscript fully available?

Reviewer #1: Yes

Reviewer #2: No

4. Is the manuscript presented in an intelligible fashion and written in standard English?

Reviewer #1: Yes

Reviewer #2: Yes

5. Review Comments to the Author

Reviewer #1: Irie-Ota et al. used LightGBM to predict postoperative visual acuity in ERM patients. The so-called "black box problem of AI" has been an issue that has hindered the widespread use of AI in the medical field, but the authors used SHAP to quantify the contribution of features to the results. The AI reported here may not directly apply to clinical practice because the examiner processes the images. However, the new approaches described above could be valuable information for readers. My minor comments are as follows.

1) Authors should state the percentage of vitrectomy combined with cataract surgery and vitrectomy alone. Is there a difference in visual acuity prediction between the two groups?

2) What is the percentage of cases where only ERM was removed or ILM and ERM were removed simultaneously? Is the visual prognosis different in the two groups?

3) Please consider adding Snellen visual acuity in the method section, as it is difficult to understand the visual acuity in decimal form.

4) “The exclusion criteria were patients with different degrees of cataracts in both eyes, a decimal best-corrected visual acuity (BCVA) of the fellow eye of <1.0.”

Does this mean that there is a cataract affecting vision? Why would Fellow eye vision less than 1.0 be excluded from the study? The meaning of this sentence is difficult to understand.

5) An explanation or citation is needed for the ERM stage in the Methods section, not the Results section.

6) The results section should be shorter. It would be easier for readers to read if some of the results were sent to supplemental data.

7) Line465. Explanation of SHAP values should be summarized in Methods.

Reviewer #2: In this study, the authors developed non-linear machine learning models to predict the postoperative visual acuity in eyes that had undergone vitrectomy for ERM. They found that the postoperative VA could be well predicted, and the area of EIFL was the most significant OCT parameter. Such study is interesting, and here are my comments:

1. In order to measure these OCT parameters, the authors must exclude those with poor quality OCT images making it difficult to determine the boundary of the different retinal layers. This means that those with DRIL must be excluded. Therefore, stage 4 ERM which represents the most severe form of ERM was excluded in this study. This must be addressed in Methods and Discussion.

2. The authors did not mention about the lens status and if any patients receiving combined vitrectomy and cataract surgery. Theoretically, phakic patients may develop vision-threatening cataract at 6 months postoperatively if they were phakic before surgery. Please add this information.

3. The significance of EIFL for postoperative VA fits clinical intuition. As the authors stated, the means that early intervention is critical in achieving favorable visual outcomes in eyes with an ERM. This should be emphasized and added in the Abstract.

4. It is difficult for clinicians to understand the meaning of Figure 4 & 5. Please revise the manuscripts to make them more understandable.

6. PLOS authors have the option to publish the peer review history of their article (what does this mean?). If published, this will include your full peer review and any attached files.

Reviewer #1: No

Reviewer #2: No

---

## [Decision Letter · Decision Letter 1]

9 May 2024

Predicting postoperative visual acuity in epiretinal membrane patients and visualization of the contribution of explanatory variables in a machine learning model

PONE-D-23-41738R1

Dear Dr. Irie-Ota,

We’re pleased to inform you that your manuscript has been judged scientifically suitable for publication and will be formally accepted for publication once it meets all outstanding technical requirements.

Kind regards,

Tatsuya Inoue

Academic Editor

PLOS ONE

Additional Editor Comments (optional):

Reviewers' comments:

Reviewer's Responses to Questions

**Comments to the Author**

1. If the authors have adequately addressed your comments raised in a previous round of review and you feel that this manuscript is now acceptable for publication, you may indicate that here to bypass the “Comments to the Author” section, enter your conflict of interest statement in the “Confidential to Editor” section, and submit your "Accept" recommendation.

Reviewer #1: All comments have been addressed

2. Is the manuscript technically sound, and do the data support the conclusions?

Reviewer #1: Yes

3. Has the statistical analysis been performed appropriately and rigorously? 

Reviewer #1: Yes

4. Have the authors made all data underlying the findings in their manuscript fully available?

Reviewer #1: Yes

5. Is the manuscript presented in an intelligible fashion and written in standard English?

Reviewer #1: Yes

6. Review Comments to the Author

Reviewer #1: The authors responded to my questions appropriately. This manuscript has significantly improved and will be of interest to readers.

7. PLOS authors have the option to publish the peer review history of their article (what does this mean?). If published, this will include your full peer review and any attached files.

Reviewer #1: No
